# Exposure of Immunogenic Tumor Antigens in Surrendered Immunity and the Significance of Autologous Tumor Cell-Based Vaccination in Precision Medicine

**DOI:** 10.3390/ijms24010147

**Published:** 2022-12-21

**Authors:** Chiao-Hsu Ke, Yi-Han Chiu, Kuo-Chin Huang, Chen-Si Lin

**Affiliations:** 1Department of Veterinary Medicine, School of Veterinary Medicine, National Taiwan University, Taipei 10617, Taiwan; 2Department of Microbiology, Soochow University, Taipei 111002, Taiwan; 3Holistic Education Center, Mackay Medical College, New Taipei City 25245, Taiwan

**Keywords:** immunogenicity, tumor antigens, immunoediting, autologous cancer vaccine, precision medicine

## Abstract

The mechanisms by which immune systems identify and destroy tumors, known as immunosurveillance, have been discussed for decades. However, several factors that lead to tumor persistence and escape from the attack of immune cells in a normal immune system have been found. In the process known as immunoediting, tumors decrease their immunogenicity and evade immunosurveillance. Furthermore, tumors exploit factors such as regulatory T cells, myeloid-derived suppressive cells, and inhibitory cytokines that avoid cytotoxic T cell (CTL) recognition. Current immunotherapies targeting tumors and their surroundings have been proposed. One such immunotherapy is autologous cancer vaccines (ACVs), which are characterized by enriched tumor antigens that can escalate specific CTL responses. Unfortunately, ACVs usually fail to activate desirable therapeutic effects, and the low immunogenicity of ACVs still needs to be elucidated. This difficulty highlights the significance of immunogenic antigens in antitumor therapies. Previous studies have shown that defective host immunity triggers tumor development by reprogramming tumor antigenic expressions. This phenomenon sheds new light on ACVs and provides a potential cue to improve the effectiveness of ACVs. Furthermore, synergistically with the ACV treatment, combinational therapy, which can reverse the suppressive tumor microenvironments, has also been widely proposed. Thus, in this review, we focus on tumor immunogenicity sculpted by the immune systems and discuss the significance and application of restructuring tumor antigens in precision medicine.

## 1. Introduction

The tumor immunoediting process includes elimination, equilibrium, and escape. The elimination phase is the same as the theory of immune surveillance, in which the immune system prevents tumor growth [1]. Tumors will be spontaneously eliminated by the immune cells, including innate [2,3] and adaptive cells [1,4,5,6,7]. In general, tumor regression can be completed in this phase. However, when only a portion of tumor cells are eliminated, the resulting tumors survive and enter the subsequent two phases, dampening the immunogenicity and facilitating tumor progression. Whether tumors remain dormant or continue to evolve during this period, the accumulation of DNA mutations or changes in gene expression still occur. These result in the modulation of tumor antigens, and the immune systems possess selective pressures by attacking susceptible tumor clones. Under selective pressures, the tumors continue to change their phenotypes and downregulate their immunogenic targets. The immune system ultimately fails to recognize and destroy tumors due to the decreased immunogenicity, so the tumors escape from the immune system and thus progressively develop [8].

The immune theory, which addresses the interaction between cancer development and host immunity, has been investigated. Thus, tumor immunotherapy is regarded as a prospective antitumor strategy and aims to strengthen the host immunity, especially cytotoxic T lymphocytes (CTLs), against tumors. The activation of antigen-specific CTL responses is triggered by antigens with high tumor specificity [5]. This highlights the significance of tumor antigens in immunotherapy. Among the immunotherapies currently used, autologous cancer vaccines (ACVs) are characterized by the expression of “individual” tumor-associated antigens (TAAs) and tumor-specific antigens (TSAs), which can escalate stronger personalized and antigen-specific immunity. Therefore, studies using antitumor vaccines with the design of individual tumor antigens are highly recommended. Furthermore, in a previous study, more cancer patients benefited from whole tumor cell-based vaccines than from specific tumor antigens [9]. This provided strong evidence for the use of whole tumor cell-based vaccines rather than immunization with the specific antigens. However, many clinical trials using ACVs have failed due to their low immunogenicity [10,11,12]. Therefore, recent studies have been focusing on increasing the effectiveness of ACVs by the induction of tumor antigens. 

The sculpting of the immunogenicity of tumors by the immune system is well understood. Tumors developed in immunocompetent hosts undergo immunoediting, which reduces the expression of TSAs and TAAs [13]. In contrast, tumors grow more aggressively in immunocompromised hosts, which increases their immunogenicity [14,15,16,17]. Furthermore, our previous studies have also indicated that tumors developed in immunocompromised hosts lead to the reprogramming of the gene and/or protein profiles with the overexpression of tumor-associated antigens [18,19]. Herein, we update the progress on the interaction between host immunity and tumor development, the impacts of host immunity on the manipulation of tumor immunogenicity, and the effective strategies of precision medicine using boosted autologous cancer vaccines. 

## 2. The Sentry of the Immune System: Immunosurveillance

### 2.1. Tumor Recognition and Rejection by the Immune System

T lymphocytes are essential in antitumor immunity, such as in the surveillance, detection, and destruction of neoplastic cells [1]. CTLs are the most potent effectors and are considered significant drivers in antitumor effects [4], and they are activated by dendritic cells (DCs). DCs initiate and maintain the antitumor T cell immunity [5]. During tumor initiation, the dying tumor cells release danger signals, such as molecules called damage-associated molecular patterns (DAMPs). Upon sensing these signals and capturing tumor cells, DCs undergo maturation, migrate to the draining lymph nodes (dLNs), and present the tumor antigens onto major histocompatibility complex I (MHC I) for presentation to CD8^+^ T cells [20]. These activate the antigen-specific CTLs, which naïve CD8^+^ T cells differentiate into CTLs and memory CD8^+^ T cells. The educated CTLs can recognize the antigenic targets expressed on the tumors and secrete cytokines, IFN-γ, perforin, and granzyme, thus performing tumor lytic functions [6,7]. In addition to priming naïve CD8^+^ T cells, DCs interact with memory T cells and induce their differentiation into the tumor sites [21]. Furthermore, DCs produce IL-12, which triggers IFN-γ release from the CTLs, which in turn enhances CTL activation and functions [22]. The close interaction between DCs and CTLs suggests tight T cell–DC cross-talk in the tumor microenvironment (TME). 

CTLs exerting antitumor effects rely on the help of other immune cells, such as DCs and helper CD4^+^ T cells. Helper CD4^+^ T cells play a prominent role in maintaining CD8^+^ cytolytic responses and preventing CTL exhaustion [4,23]. Interestingly, the help signals originating from the CD40 ligand (CD40L) on the helper CD4^+^ T cells stimulate CD40 on the DCs, which deliver CD4^+^ T cell-derived help signals to CD8^+^ T cells and elicit CTL responses [24]. During the first antigen priming step, naïve CD4^+^ T cells and CD8^+^ T cells are separately activated by different populations of DCs in the peripheral lymph nodes [25,26,27], where MHC I-expressing cDC1 usually interacts with CD8^+^ T cells. In contrast, MHC II-expressing cDC2 interacts with CD4^+^ T cells. Next, in the second priming step, CD4^+^ T cells and CD8^+^ T cells interact with the same cDC1, and the help signal occurs [28,29]. cDC1 engages in cognate interaction with pre-activated CD4^+^ T cells, which optimizes the cDC1 to relay signals for the differentiation of effector T cells and memory CTLs to pre-activated CD8^+^ T cells [30]. In summary, antigen-specific (pre-activated) CD4^+^ T cells assist the pre-activated CD8^+^ T cells in the differentiation into effector T or memory T cells. Helper CD4^+^ T cells dictate the quality of the CTL differentiation and promote the expansion of antigen-specific CTLs by cytokine signals. CD4^+^ T cells promote the development of antigen-specific CTLs through the amplification of IL-12 and IL-15 production induced by IFN-γ in DCs [31,32]. Furthermore, IL-2 signaling promotes CD8 proliferation [33,34], and the expression of IL-2 receptor α-chain (IL-2Rα) by recently primed CD8^+^ T cells depends on CD4^+^ T cells [35]. In addition to the assistant roles of CD4^+^ T cells, the cytolytic activities in antitumor effects of CD4^+^ T cells have been proposed. CD4^+^ T cells produce IFN-γ and express granzymes and perforin to kill tumors [36,37]. Recent studies using RNA sequencing in human cancers have reported cytotoxic characteristics of CD4^+^ T cells, which express cytolytic molecules such as granzymes, perforin, and granulysin in the tumors and circulation of cancer patients [38,39,40,41]. 

Natural killer (NK) cells are also essential effectors in tumor immunosurveillance. For example, NK cells can be activated by MHC class I polypeptide-related sequence A (MICA) and MICB, which are expressed on tumors [2]. NK cells directly kill tumor cells by releasing perforin and granzymes [3] or trigger cell apoptosis through the ligation of cell death receptor-mediated pathways (FasL/Fas) [42]. Healthy cells express high levels of MHC I, which ligates to the killer immunoglobulin-like family inhibitory receptors (KIRs) on NK cells. However, tumors will downregulate MHC I expression to evade CTL-mediated cytotoxicity and simultaneously activate NK cells due to the decreased inhibitory signaling [42,43] (Figure 1). 

### 2.2. Failure of Immunosurveillance Enables Cancer Progression

Accumulating evidence indicates the increased risk of tumor development in immunosuppressed patients, suggesting the crucial antitumor role of intact immunity. Mice with a deficiency in adaptive immunity provide a practical animal model for testing cancer immunosurveillance. In one study, mice with severe combined immunodeficiency (SCID) exhibited impaired differentiation of both T and B lymphocytes, and thus 15% of these mice developed T cell lymphomas [44]. In another study, *RAG2*^−/−^ mice, another kind of mice without B and T cells, developed spontaneous intestinal adenomas (50%), intestinal adenocarcinomas (35%), and lung cancers (15%) at the age of 15–16 months [45]. In addition to adaptive immunity, NK cells also manifested cancer immunoediting by producing IFN-γ and induced M1 macrophages. 

Mice with a deficiency in effector cells, such as T cells and NK cells, usually develop spontaneous cancers with aging. In STAT1-deficient mice, the JAK/STAT1 signaling pathway is down-regulated and thus inhibits type I and II IFN production. This leads to the increased formation of mammary carcinomas [46]. Mice lacking death-inducing molecule tumor necrosis factor (TNF)-related apoptosis-inducing ligand (TRAIL) [47], or with the inactivation of the Fas-mediated cell death pathway by Fas or FasL mutation, have accelerated hematological malignancies [48]. Decreased cytokine production and/or antigen presentation in gene-deficient mice, including *Perforin*^−/−^ (lack perforin) [49], *Ifng*^−/−^ (lack IFN-γ) [50], *Perforin*^−/−^*Ifng*^−/−^ (lack perforin and IFN-γ) [50], *Perforin*^−/−^*B2m*^−/−^ (lack perforin and MHC class I expression) [51], and *Lmp2*^−/−^ (defective MHC class I antigen presentation) [52], facilitates tumor growth. Due to the decreased IFN-γ secretion, IL-12 and IL-18, two IFN-γ-inducing cytokines, are down-regulated. A previous study found that, compared with wild-type mice, neither IL-12 nor IL-18 deficient mice exhibited increased incidence of tumor development [50]. These results suggest that immune effector cells, cytokines, and their related pathways are essential for the regression of tumor development in immunosurveillance. The following findings support the same conclusion. Due to the presence of the constitutively expressing oncogene, Kras, and inhibiting tumor suppressor, p53, tumors growing in immunocompetent mice were retarded compared to those in immunocompromised mice.

## 3. Immunoediting: How Tumors Hijack Host Immunity and Establish a Favorable Microenvironment

Tumors escape from the immune system via several mechanisms, including the formation of regulatory cells, reduced immune recognition, and production of suppressive cytokines. The immune system fails to restrict tumor development; as a result, tumors evade immune recognition (due to diminished tumor antigens and immunogenicity) and release suppressive cytokines. The regulatory immune cells, Tregs, MDSCs, dysfunctional DCs, and M2 macrophages also express immune regulatory factors, such as IDO and arginase, to construct a tumor microenvironment (TME) that enhances tumor progression and dampens T cell functions. In this escape phase, the balance is toward tumor progression, with the infiltrations of inhibitory cells, cytokines, and factors. The results are that the immune system is incapable of inhibiting tumor progression. By generating the appropriate TME via the mechanisms listed below, tumors dampen the immune responses, supporting tumor growth and even metastasis [8] (Figure 2). 

### 3.1. Recruitment of Regulatory Cells

#### 3.1.1. Regulatory T Cells

Immune suppression mediated by the regulatory cells and other suppressive factors in the TME is a major mechanism by which tumors avoid attacks from the immune system. CD4^+^CD25^+^FoxP3^+^ tumor-derived regulatory T cells (Tregs) play central roles in immune suppression. Studies also show that different cytokines, such as IL-10 and TGF-β produced by the tumors, trigger the conversion of CD4^+^ T cells into suppressive Tregs [53]. Tregs then secret IL-10, TGF-β, and IL-35 to downregulate antitumor immunity, suppress the antigen presentation by DCs, and decrease the tumor-specific CTLs [54]. Meanwhile, since IL-2 is essential to T cell activation and maintenance, these regulatory T cells compete with effector T cells by largely consuming IL-2 [33,34]. Tregs can also kill the CTLs by the secretion of perforin and granzyme, leading to osmotic lysis and apoptosis, just as CTLs and NK cells do to tumor cells. Furthermore, Tregs interfere with the memory T cells by repressing their effector and proliferation activities through the upregulation of CTLA-4 ligands [55]. The aforementioned evidence supports the mediation of the comprehensive suppression of antitumor immunity by Tregs.

#### 3.1.2. Myeloid-Derived Suppressive Cells

Both Tregs and myeloid-derived suppressive cells (MDSCs) facilitate their own expansion by the overexpression of CD73 [56,57], TGF-β [58], and indoleamine 2, 3-dioxygenase (IDO) [59,60,61]. In addition, MDSCs may promote the recruitment of Tregs by producing CCR5 ligands, CCL3, CCL4, and CCL5 [62]. MDSCs express a high level of inducible nitric oxide (iNO), which produces nitric oxide (NO) [63,64]. NO inhibits the JAK/STAT5 pathway and/or suppresses the antigen presentation from DCs, leading to the suppressive proliferation of effector T cells [64,65]. Furthermore, MDSC-derived NO also triggers the effector T cell apoptosis [66] and reduces E-selectin expression on endothelial cells, which hampers the migration of T cells to the tumor sites [67,68]. MDSCs also secrete a high level of arginase 1 (ARG1), which promotes Treg expansion [69] and depletes the L-arginine, a conditionally essential amino acid in effector T cell functions. These actions lead to the dysregulation of effector T cells and the propagation of Tregs [70].

#### 3.1.3. Tumor-Associated Macrophages

Among the regulatory cell types, tumor-associated macrophages (TAMs) are the most abundant population in TME [71]. There are two polarization states of TAMs: classically activated M1 and alternatively activated M2 subtypes [72]. M2 macrophages produce several anti-inflammatory cytokines, such as IL-4, IL-10, IL-13, vascular endothelial growth factor (VEGF), and TGF-β, to inhibit immune systems and promote tumor progression [73,74,75]. Therefore, CD11b^+^F4/80+ macrophages having an M2 phenotype are considered regulatory (or “bad”) macrophages [76], which are essential for tumor growth and metastasis. [77]. Furthermore, M2 macrophages can support tumor-related vasculature by accumulating vascular endothelial cells [78]. M2 polarization is induced by several factors, one of which is a colony-stimulating factor 1 (CSF1). CSF1 plays a fundamental role in pro-angiogenesis and tumor burden increase [79], revealing M2 macrophages as the efficient cancer development enabler within the TME [77]. In fact, M2 participates in the angiogenesis cascade, which consists of a series of pro-tumor functions, including degradation of extracellular matrix (ECM), endothelial cell proliferation, and migration [80]. M2-related enzymes promote ECM deposition and the proteolysis of collagens. The degraded collagen fragments may stimulate M2, which could further re-arrange stroma and enhance the angiogenesis activities [81]. Therefore, M2 macrophages are immune cells that promote tumors by releasing suppressive cytokines, triggering tumor angiogenesis, and reorganizing the ECM in the TME.

### 3.2. Defective Antigen Presentation

#### 3.2.1. Manipulation of the DC Lineage

Defective antigen presentation is another fundamental mechanism by which tumors evade immune surveillance. Immunogenic tumors will be recognized and destroyed by immunosurveillance. However, tumors evolve multiple skills for escaping from immune recognition, including the suppression of DC functions and downregulation of MHC-I expression [82]. DCs play essential roles in the initiation of antitumor T cell immunity [5]. Among the DC subsets, cDC1s (c: conventional) are the most important [83], for the abundance of cDC1s in the TME is associated with T cell infiltration and overall survival in cancer patients [20,84]. cDC1s are recruited to tumor regions by chemokines released by the tumors, such as CCL4 and CCL5 [85,86]. After taking up tumor cells, cDC1s will mature in the tumor sites and migrate to the dLNs to process tumor antigens onto MHC-I for presentation to CD8^+^ T cells [20]. These processes result in the activation of antigen-specific CTLs. Thus, cDC1s participate in antitumor activities and serve as targets for tumors to escape from the immune system. Tumors prevent the accumulation of cDC1s by activating their β-catenin signaling pathways, thereby decreasing the infiltration of cDC1s and T cells [85,86]. Furthermore, tumors release high levels of prostaglandin E_2_ (PGE_2_) [87], VEGF [88,89], IL-6 [90], IL-10 [91], and TGF-β [92] to suppress DC maturation and differentiation. Thus, tumors can decrease their antigenicity by manipulating DC functions, which become tolerogenic and immunosuppressive phenotypes.

#### 3.2.2. Sabotage of the Machinery of Antigen Presentation

The downregulation of tumor antigenicity, such as MHC class I loss, is a well-studied characteristic of immune evasion from tumor-specific cytolytic effects. Tumor antigenicity is sculpted during immunoediting in an immunocompetent host. Some of these changes include antigen depletion [93], gene alterations in *MHC-I* and *B2M* [94], and modulation of other antigen processing and presentation machinery (APM). Several TAAs are the by-products during tumorigenesis, which indicates that these TAAs are not necessarily functional for tumor growth. The loss of such antigens in tumors can prevent immune predation [82]. *MHC-I* and *B2M* mutations are usually found in tumors [94], and they lead to reduced surface expression of MHC-I [95]. Diminished B2M expression is correlated to a cold immune environment, with low T cell infiltration [93]. Tumors evading immune surveillance by downregulation of APM components are also widely described. Deficiency in proteasome subunits [96], transporters associated with antigen processing (TAP) proteins [97], and Tapasin [98] result in inadequate antigen presentation, thereby escaping CTL recognition. In conclusion, the downregulation of tumor antigens and antigen-presenting processes leads to enhanced tumor growth and metastasis as the CTLs fail to identify the targets on the tumor cells. 

### 3.3. Immune Suppressive Mediators

Tumors can also inhibit host immunity by releasing regulatory cytokines. Within the TME, tumor-induced cytokines and inflammation facilitate cancer development [99,100]. Several tumor-promoting cytokines are regulated by nuclear factor-κB (NF-κB), a central orchestrator of inflammation. Previous research has revealed that the inactivation of NF-κB in immune cells decreases the expression of proinflammatory cytokines and thus relieves the tumor burden [101]. These results indicate that NF-κB-mediated cytokines, including IL-1 [102], IL-6 [103], IL-8 [104], and TNF-α [102], are highly correlated to tumorigenesis. For example, IL-6 interacts with the receptor JAK and induces STAT-3 activation [103], which triggers oncogenes such as myeloid cell leukemia-1 (MCL-1) and upregulates proliferative genes, Cyclin-D1, in tumor cells [105]. In addition, TGF-β is one of the most important contributors to tumor growth. TGF-β is released by the cancer cells, Tregs, fibroblasts, and other types of cells within the TME. The elevated TGF-β levels inhibit effector T cell differentiation, promote Treg proliferation, and dampen DC functions [106]. Furthermore, TGF-β enhances the epithelial-to-mesenchymal transition (EMT), synergistically triggering tumor metastasis with IL-6 via the overactivation of JAK/STAT signaling pathways [107]. Therefore, TGF-β is usually regarded as a chief mediator within the immunosuppressive factors [108]. 

The overproduction of PGE_2_ [87], VEGF [88,89], IL-6 [90], IL-10 [91], and TGF-β [92] from tumors will inhibit DC differentiation. The undifferentiated DCs fail to appropriately present antigens and are unable to educate T cells. Immunosuppressive enzymes, such as IDO and arginase, also cause tumor progression through the induction of T-cell tolerance and tumor cell proliferation. IDO impairs CTL functions through the downregulation of the T cell receptor ζ chain and enhances Treg generation [109,110]. Elevated intertumoral IDO expression is correlated to T cell dysfunction, such as decreased levels of granzyme B in tumor-infiltrating CD8^+^ T cells [111], impaired degranulation of γδ T cells [112], and upregulation of PD-1 and PD-L1 ligands [113,114]. Tumor-infiltrating myeloid cells produce high levels of arginase, which inhibit T cell receptor expression, dampen antigen-specific T cell antitumor immunity, and induce Treg proliferation [115,116]. 

### 3.4. Deletion of Tumor-Specific CTLs

Tumors and several immunosuppressive cells trigger T cell apoptosis by Fas and Fas ligand (FasL) signaling pathways. FasL is expressed by tumors and can induce apoptosis of Fas-expressing antitumor CTLs [117]. FasL is reported to be expressed on the tumoral endothelium but not normal vasculature [118], suggesting that FasL-expressing endothelial cells in tumors induce Fas-mediated apoptosis in CTLs. Furthermore, these phenomena have only been observed in CTLs, not Tregs [119,120]. FasL-expressing MDSCs also result in the deletion of CTLs [121,122]. 

The high expression of CD70 and PD-L1 on the surfaces of tumors also mediates T cell death. The TNF receptor family member CD27 is usually expressed on T cells [123], but its ligand, CD70, is overexpressed in some tumors [124]. The dysregulation of the CD70–CD27 axis within the TME is associated with immunosuppression [125]. One explanation is that the over-activation of CD27 can lead to T-cell deletion, for Siva, a pro-apoptotic protein, can interact with CD27 through a caspase-dependent pathway [126]. Therefore, tumors can induce immunosuppression by promoting T-cell apoptosis via the expression of CD70.

PD-L1 and PD-1 induce immune suppression and facilitate tumor growth by induction of T cell apoptosis. PD-L1 is highly expressed in numerous types of cancers, including numerous solid tumors and hematological cancers [127]. When tumor-expressing PD-L1 combines with PD-1 on the T cells, SHP-1/2 are recruited to the C-terminal PD-1. This causes the de-phosphorylation of several vital signal transducers, including ZAP70, CD3δ, and PI3K pathways, and thus inhibits T cell proliferation, reduces cytokine production, and triggers T cell apoptosis [127,128]. 

## 4. Greater Immunogenicity of Tumors Developed in Immunodeficient Hosts 

### 4.1. Increased Immunogenicity of Tumors Developed in Immunocompromised Hosts (Figure 3)

Genomic instability, such as mutation, is one of the essential features of tumors [129]. The uncontrolled growth in tumors ultimately results in alterations of tumor antigens, some of which may harbor high immunogenicity [17]. A previous study showed that immunodeficient (RAG2^−/−^x γc^−/−^) mice were more susceptible to MCA (3-methylcholanthrene)-induced sarcoma than were syngeneic RAG2^−/−^ and wide-type (WT) mice. Furthermore, the authors harvested the tumors from these three strains of mice and re-inoculated the primary tumors into immunocompetent mice. MCA-induced sarcoma cell lines derived from RAG2^−/−^ (30% regression) and RAG2^−/−^x γc^−/−^ (70% regression) mice failed to form tumors when transplanted into syngeneic WT mice. In contrast, all the MCA-induced sarcoma cell lines from WT mice successfully developed when re-inoculated into immunocompetent mice. These results suggest that tumors derived from an immunodeficient host are more malignant with the increasing tumor antigens. Therefore, with their high immunogenicity, tumors derived from RAG2^−/−^x γc^−/−^ mice attract the incremental infiltration of immune cells, followed by tumor remission in WT mice [14]. 

Tumor immunoediting is the consequence of immunoselection; this is especially true for the T-cell-dominant responses, which lead to the generation or tumors that exhibit reduced immunogenicity [13,15]. In contrast, tumors developed in immunodeficient hosts will not undergo the selective process from immune cells, so they can develop more aggressively with a higher tumor mutational burden (TMB). Schreiber et al. [130] reported that tumors in RAG2-deficient mice were more immunogenic than those in immunocompetent hosts. Furthermore, exome sequence analysis was conducted to determine the mutational landscape of highly immunogenic (unedited) tumors derived from RAG2^−/−^ mice. Unlike the WT mice, the RAG2^−/−^ mice had a point mutation in Spectrin- β2, which was validated as the source of a neo-epitope in tumors grown in RAG2^−/−^ mice [15]. These results reveal that desirable antigens and/or increased immunogenicity will be generated when tumors develop in immunodeficient hosts [16]. 

**Figure 3 ijms-24-00147-f003:**
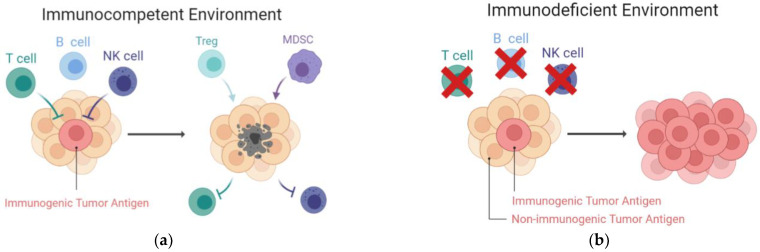
Increased immunogenicity of tumors developed in immunodeficient hosts. (**a**) In immunocompetent hosts, tumors with high immunogenicity will be spontaneously destroyed by the T cells and NK cells. The resulting tumors, which express low immunogenicity, survive and create a suppressive TME. Within the TME, inhibitory cells, such as Tregs and MDSCs, facilitate tumor progression and deplete the effector cells; (**b**) In contrast, tumors established in an immunocompromised host are more malignant and immunogenic. Without the selective pressures from T and NK cells, antigenic tumors are retained. MDSCs, myeloid-derived suppressive cells; NK cells, natural killer cells; TME, tumor microenvironment; Tregs, regulatory T cells.

### 4.2. Differentially Expressed Profiles in Tumors Developed in Immunodeficient Hosts 

According to the immunoediting theory, tumors can alter the antigen expression profiles to escape the hunt for host immunity. Our previous findings have provided significant evidence supporting this theory. The canine transmissible venereal tumor (CTVT) is one of the few spontaneously occurring cancer models for investigating the interaction between tumor development and the host immune system [131]. Though well-proven not to be induced by a virus infection, CTVT is a contagious tumor that can break the MHC barriers and is transmissible among dogs. This phenomenon reveals the strong living instinct of this tumor, but it remains a problem to explain this characteristic. So far, CTVT cannot be prepared as a cell line, and the inoculation of this tumor into dogs or immunodeficient mice is the only solution for CTVT maintenance. By chance, we observed the rapid growth pattern of CTVT that had previously been inoculated into NOD.CB17-Prkdcscid/NcrCrl (NOD.SCID) mice for live tumor conservation (named as XCTVT, X: xenogeneic). XCTVT developed in dogs showed an excellent proliferation ability and even metastasized in some cases, which seldom occurs in the original CTVT. We then analyzed the whole transcriptomes between the parent CTVT and CTVT once sojourned in NOD.SCID mice (named as MCTVT, M: mice). The result show that the gene expression profiles significantly differed between the two tumors. In MCTVT, the mRNA involved in the dysregulated interaction of chronic inflammation, chemotaxis, and extracellular space modification are highly transcribed, which implies their roles in promoting cancer progression [19]. This raised the question of whether the altered tumor antigenic topology in CTVT once seeded in NOD.SCID mice were just an exception, or whether the competent versus defective host immunity actually did have impacts on the profiles of TAAs. We then verified this issue using the colorectal carcinoma model, known as CT26 tumors, and similar results were found. Comprehensive analysis of protein profiles from CT26 tumors (wild type) and CT26 tumors developed in NOD.SCID mice (CT26/SCID) by liquid chromatography/mass spectrometry (LC-MS/MS) revealed that CT26 cells inoculated on NOD.SCID mice display more than 400 significantly differential expression proteins (DEPs). Several novel therapeutic targets were identified as overexpressed on CT26/SCID tumors, including KNG1, apoA-I, and β2-GPI, which could effectively activate cytotoxic T cells and directly suppress tumor proliferation [18]. 

### 4.3. Increased Antigenicity of Tumors Developed in Immunocompromised Hosts 

The DEP landscapes shown in the CT26/SCID model have proven the quantitative modulation or qualitative alteration of the antigen repertoire presented on tumors. A variety of TAAs have bloomed on CT26/SCID cells because the tumor development within immunodeficient hosts may relieve the microenvironmental stress to tumors and reshape the TAA expression profiles. The customized TAAs from tumors that have ever been through the immunocompromised host immunity may evolve to produce heterogenic tumor antigens and operatively trigger antitumor immunity. To clarify the immune-stimulatory effects of these TAAs expressed in CT26/SCID, we prepared the proteins from CT26/SCID tumors and vaccinated CT26 tumor-bearing BALB/c mice. Compared with CT26/WT, CT26/SCID tumor lysates could significantly suppress tumor growth, increase CD3^+^ tumor-infiltrating lymphocytes (TILs), enhance CTL functions, and trigger Th-1 predominant immune responses. These T cells could produce IFN-𝛾 and possessed specific tumor cytotoxicity [18]. Therefore, tumors inoculated into hosts with defective immunity could reprogram the tumor antigenic expressions. Harvesting these cells as the major component of cancer vaccines could be a promising and effective strategy. 

## 5. The Era of Tumor Antigens in Immunotherapies 

### 5.1. Tumor-Associated Antigens 

Tumor antigens recognized by the immune system can be classified by their tumor specificity. TSAs are only found in cancer cells, and TAAs are variably expressed in normal cells [132]. TSAs, such as mutation-derived antigens, are distinguishable from normal cells. For example, basal and Her2-positive tumors contain more mutated proteins and TP53 mutations than luminal A/B breast tumors do [133]. Mutations of β-catenin [134], caspase-8 [135], KRAS [136], and BCR-ABL [137] have been found in different tumors, and they could be effective therapeutic targets for cancer management. The aberrant expression of TAAs on the tumors also makes them targets for antitumor immunity. RAGE-1 [138], MUC-1 [139], and Her2/neu [140] are well-known TAAs and have been applied in cancer vaccines. Immunogenic antigens have high potential for immunotherapy, as these antigens can be captured by DCs, thereby triggering T-cell recruitment to the tumor sites [141]. The well-educated CTLs can release IFN-𝛾, perforin, and TNF-α, thus precisely eradicating the tumors [142]. Though TAAs are presumed to induce a minor immune response, which is insufficient for tumor rejection, these antigens are essential factors in the development of immunotherapy agents. 

### 5.2. Differentially Expressed Profiles in Tumors Developed in Immunodeficient Hosts 

Tumors are heterogeneous populations containing tumor cells and the surrounding microenvironment, such as fibroblasts, endothelial cells, Tregs, TAMs, MDSCs, and the ECM. These cells and factors help create an immunosuppressive TME leading to tumorigenesis, tumor progression, and metastasis. Stroma cells are more genetically stable than tumor cells [143]. Furthermore, tumors will decrease immunogenicity by several mechanisms to escape CTL recognition. These characteristics make stroma cells potential candidates for therapeutic targets. 

Tumor stroma cells display physiological functions distinct from their normal counterparts under the impacts of TME, and they exhibit elevated tumor stroma-associated antigens (TSAAs). Well-known tumor stromal cells are cancer-associated fibroblasts (CAFs), tumor endothelial cells (TECs), and TAMs, all of which have been proposed as possible cellular targets in cancer therapies [144]. As the most abundant cell type in TME, CAFs secrete growth and angiogenic factors that remodel the tumor ECM, inhibit the antitumor immunity, and support tumor cell proliferation [143,144]. Furthermore, CAFs overproduce fibroblast activation protein α (FAP-α), a surface glycoprotein selectively expressed on the stroma of numerous solid tumors [145,146], to enable tumor growth, angiogenesis, and metastasis [147,148]. These known findings make TSAA-expressing stromal cells ideal therapeutic targets for cancer treatments. 

### 5.3. Tumor Cell-Based Vaccine in Current Antitumor Strategies 

Targeting both tumor cells and stroma is considered a promising strategy to remove tumors [149]. The aforementioned findings have spurred the development of autologous tumor cell-based vaccines. Autologous tumor cell-based vaccines can be prepared by different methods, and their immunogenicity can be improved through various approaches. Whole tumor cell lysates are usually prepared by homogenization [18], UV-ray irradiation [150], repeated cycles of freezing and thawing [151], and hyperthermia [152,153,154]. Cancer vaccines processed by homogenization express the total tumor proteins, preserving personalized TAAs, TSAs, and antigenic elements from the surrounding stroma. These vaccines can induce robust antigen-specific immunity against tumors and stromal cells [155]. The UV-treated tumor lysates induce apoptosis in tumor cells, and the exposure of phosphatidylserine (PS) on the surface of tumors facilitates the uptake and cross-presentation by DCs [150]. The apoptotic cells release high mobility group box1 (HMGB1) [156] and pentraxin-3 (PTX3) [157] and thus induce DC maturation. Tumor cell lysates prepared by a repeated freeze–thaw process undergo necrosis, and the necrotic cells can induce DC maturation [151]. Hyperthermia is an effective method of manufacturing tumor vaccines and improving the immunogenicity of tumors [154]. After being heated, tumors enhance their surface expression of MHC I-related molecules [153] and the antigens, leading to higher cytolytic effects and antigen-specific CD8^+^ T cell accumulation [152]. 

Therapeutic tumor cell-based vaccines rely on the concept of expressing “unique” and “personalized” TAAs, which can trigger more cancer-specific CTLs. As previously described, tumors are heterogeneous, and multiple treatments simultaneously inhibiting tumor proliferation and reversing the suppressive TME are promising antitumor strategies. Therefore, numerous clinical trials using cancer vaccines in combination with other therapies have been widely reported (Table 1). For example, the Her2/neu cancer vaccines combined with granulocyte macrophage-colony stimulating factor (GM-CSF) significantly increased the antigen-specific T responses and prolonged the disease-free survival (DFS) in patients with triple-negative breast cancers [158]. Her2-targeted cancer vaccines with trastuzumab treatments improved the DFS in high-risk breast cancer patients [159]. In advanced melanoma cases, the gp100 cancer vaccine combined with IL-2 therapy significantly improved the median survival in patients as compared with the use of IL-2 alone [160,161]. Furthermore, combining cancer vaccines with chemotherapy reduced myeloid suppressive cells and enhanced the immune responses in advanced cervical cancer patients [162]. Radiotherapy with cancer vaccination induced elevated infiltrations of vaccine-induced CD8^+^ T cell responses in neck cancer patients [163]. Anti-angiogenic molecules, which reverse the suppressive TME, synergistically enhance the therapeutic effects of cancer vaccines [164]. Antagonists of inhibitory receptors or agonists of co-stimulatory molecules are widely proposed as a combinational therapy with cancer vaccines. Anti-CTLA-4, anti-PD-1, anti-PD-L1, and other blockades of inhibitory receptors (Lag-3 and Tim-3) have shown synergy with cancer vaccines [165,166,167]. Co-stimulatory molecules, including CD40 [168], OX40 [169], and 4-1BB [170] antibodies, have been used in combination with cancer vaccines to augment cancer vaccine efficacy by increasing immunogenicity. These successful clinical trials have shown that both the cancer-specific immunity induced by tumor antigens and the factors that are able to effectively reverse the immunosuppressive TME are both important in tumor control. 

## 6. Conclusions

It is well proven that tumors developed in defective immunity can stimulate tumor antigenic expressions. Therefore, harvesting these cells as the major component of ACVs could effectively boost immunogenicity and antitumor efficacy (Figure 4). Furthermore, combinational immunotherapy has benefited cancer patients greatly and been viewed as a prospective choice for cancer treatments. To date, many immune-boosting strategies can be used in combination with ACVs. The blockade pathways (CTLA-4, CD80, CD86, PD-1, PD-L1), co-stimulatory signaling (CD40, OX40, 4-1BB), cytokines (IL-2, GM-CSF), chemotherapy, radiation, or any strategies that can reverse the suppressive TME are all potential options. The optimization of ACVs and collaborative treatments will lead to future clinical success.

## Figures and Tables

**Figure 1 ijms-24-00147-f001:**
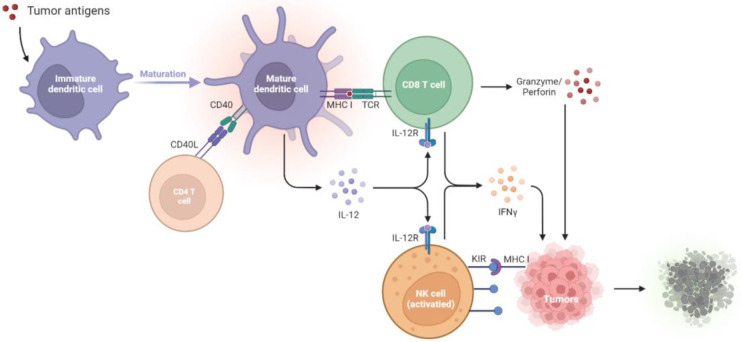
Tumor suppression by the immune systems during immunosurveillance. Once tumor antigens are captured, immature DCs undergo maturation. The mature DCs present the antigens onto MHC I molecules for presentation to CD8^+^ T cells. The CD40L on the CD4^+^ T cell stimulates CD40 on the DCs, which delivers help signals to CD8^+^ T cells. With the CD40–CD40L interaction, the amplification of IL-12 in DCs promotes the development of antigen-specific CD8^+^ T cells and thus increases the IFN-γ, granzyme, and perforin production to kill tumors. NK cells will be activated by the diminished expression of MHC I on tumors, which relieves the KIR inhibitory signaling and activates the NK cell toxicity towards tumors. CD40L, CD40 ligand; DCs, dendritic cells; IL-12R, IL-12 receptor; IFN-γ, interferon-γ; KIR, killer immunoglobulin-like receptor; MHC I, major histocompatibility complex class I; NK cells, natural killer cells; TCR, T cell receptor.

**Figure 2 ijms-24-00147-f002:**
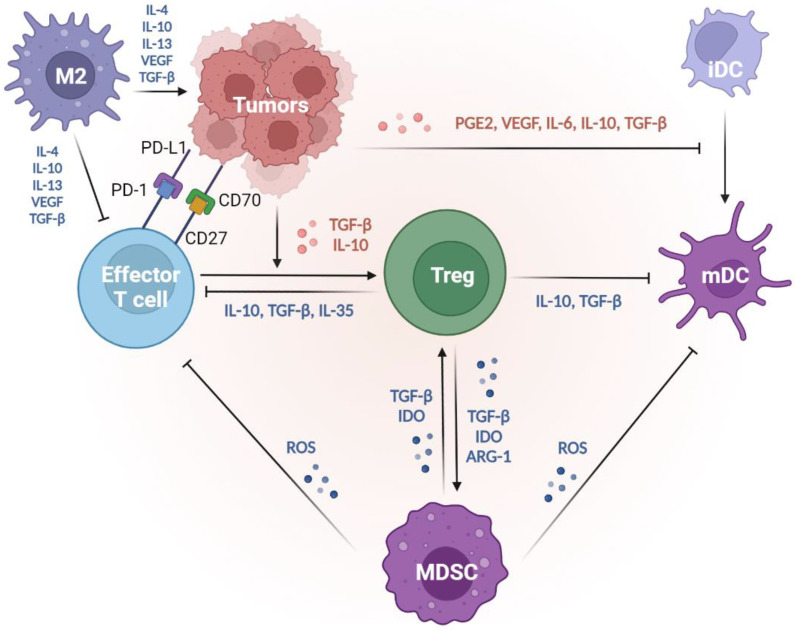
Suppressive tumor microenvironment created by tumors, thus decreasing tumor immunogenicity and evading immunosurveillance. Tumors suppress the DC functions to reduce their antigenicity. With the impacts of PGE_2_, VEGF, IL-6, IL-10, and TGF-β produced by tumors, immature DCs fail to undergo maturation, so the antigen presentation abilities are suppressed. Tumors evade the attack from T effectors from several mechanisms. First, tumors mediate T-cell apoptosis via dysregulation of the CD70–CD25 axis and the ligation of PD-L1 and PD-1. Second, tumors produce IL-10 and TGF-β to trigger effector T cells into regulatory phenotypes (Treg), which play central roles in immune suppression. Through the release of IL-10, TGF-β, and/or IL-35, Tregs dampen the effector T cell activities and suppress the antigen presentation of mature DCs. Furthermore, factors such as TGF-β, IDO, and ARG-1 produced by both Tregs and/or MDSCs shape a positive feedback correlation to facilitate the expansion of each group and strengthen the suppressive TME. MDSCs secrete a high level of ROS, suppressing the antigen presentation from mature DCs and the anti-tumor immunity from effector T cells. In the TME, M2 macrophages are the most abundant tolerogenic cells. M2 macrophages produce IL-4, IL-10, IL-13, VEGF, and TGF-β to inhibit effector T cells and promote tumor progression. Thus, tumors can decrease their antigenicity by manipulating DC functions and inhibiting effector T cells, thereby evading immunosurveillance. ARG-1, arginase 1; DCs, dendritic cells; IDO, indoleamine 2, 3-dioxygenase; MDSCs, myeloid-derived suppressive cells; PGE_2_, prostaglandin E_2_; TGF-β, transforming growth factor-β; TME, tumor microenvironment; Tregs, regulatory T cells; VEGF, vascular endothelial growth factor.

**Figure 4 ijms-24-00147-f004:**
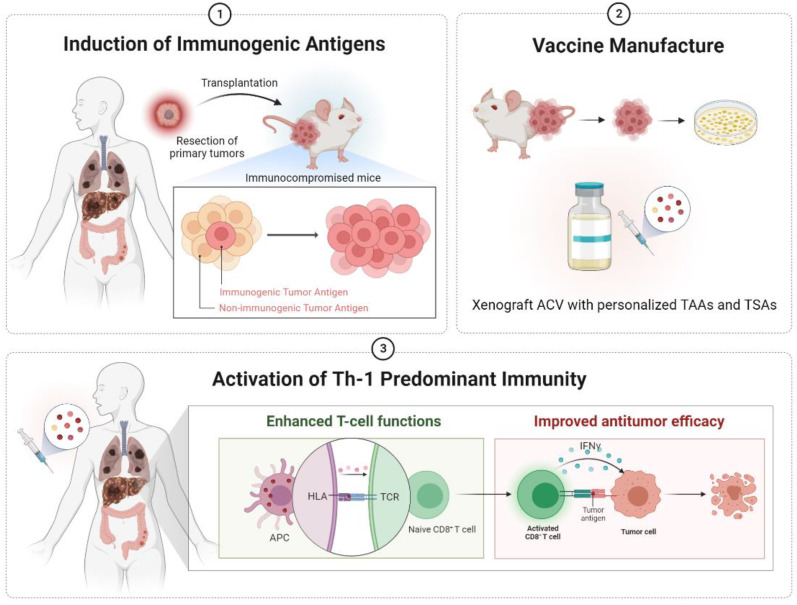
Diagrams illustrating the hypothesis in the current study. Tumors developed in a defective immune environment trigger the overexpression of immunogenic antigens. Based on this theory, we hoped to inoculate a few primary tumors from clinical cancer patients into immunocompromised mice. Then, the tumors progressively developed with the generation of personalized tumor antigens. The tumors were harvested and manufactured into xenograft ACVs. With the xenograft cancer vaccination, the APCs could recognize the unique tumor antigens and educate the naïve CD8^+^ T cells. The activated (educated) CD8^+^ T cells exerted cytolytic effects by releasing cytokines, such as IFN-γ. ACV, autologous cancer vaccine; APC, antigen-presenting cells; HLA, human leukocyte antigen; IFN-γ, interferon-gamma; TAAs, tumor-associated antigens; TCR, T-cell receptor; TSAs, tumor-specific antigens.

**Table 1 ijms-24-00147-t001:** Selected cancer cell-based vaccines achieved to clinical trials. DCs, dendritic cells; PFS, progression-free survival; GM-CSF, granulocyte macrophage-colony stimulating factor; TNBC, triple negative breast cancer; DFS, disease-free survival; RT, radiotherapy.

Phase	Interventions	Indication	Molecular Target	Efficacy	NCT Number
Tumor cell-derived RNA vaccines					
Phase I/II	Autologous DCs transfected with patients’ tumor mRNAs	Glioblastoma	Autologous cancer stem cell mRNA	Safe, well-tolerated, and prolonged PFS	NCT00846456
Phase I/II	Autologous DCs transfected with patients’ tumor mRNA	Advanced malignant melanoma	Complete autologous tumor-mRNA	Safe and detectable T-cell responses in about 50% of the patients	NCT01278940
Personalized mutation-based vaccines					
Phase I/Ib	Combined with RT (approximately 60 Gy over 6 weeks)	Glioblastoma	Synthesized personalized neo-peptides	Safe and increased neoantigen-specific T cell responses	NCT02287428
Phase I	Combined with poly-ICLC	Melanoma	Synthesized personalized neo-peptides	Safe, increased T cell responses, and long-term persistence of neoantigen-specific T cells	NCT01970358
Phase I	Combined with or without treatment with RBL001/RBL002	Stage III or IV malignant melanoma	10 potentially immunogenic mutated sequences per patient	Safe and well-tolerated	NCT02035956
Tumor-associated antigen vaccines					
Phase IIb	Combined with GM-CSF	Breast cancer (including TNBC)	Her2/Neu protein (nelipepimut-S)	Safe and prolonged DFS in TNBC patients	NCT01570036
Phase III	Combined with IL-2	Stage III or IV cutaneous melanoma	Gp100	Safe, well-tolerated, and prolonged DFS	NCT00019682

## Data Availability

Not applicable.

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
