# Peer review of "Exposure of Immunogenic Tumor Antigens in Surrendered Immunity and the Significance of Autologous Tumor Cell-Based Vaccination in Precision Medicine"

_ijms, 2022, doi:10.3390/ijms24010147_

Round 1

Reviewer 1 Report

In the manuscript entitled “Exposure of immunogenic tumor antigens in surrendered immunity and the significance of autologous tumor cell-based vaccination in precision medicine” the authors discuss the utility of autologous cancer vaccines, the reprogramming of tumor antigenic expression, combinational anti-tumor therapies, and their relevance to the precision medicine. Below are the comments to improve the manuscript.

1.    Lines# 206-207: CD4+CD25+FoxP3+ tumor-derived regulatory T cells (Tregs) play central roles in immune suppression.  The + sign denoting the expression of markers should be superscripts throughout the manuscript.

2.    Lines# 386-388: CB17-Prk- 386 dcscid/NcrCrl (NOD.SCID) mice for live tumor conservation (named as XCTVT, X: xenogeneic). A thorough spelling check should be carried out throughout the manuscript.

3.    The authors should include diagrams to depict their take-home message/hypothesis.

Author Response

In the manuscript entitled “Exposure of immunogenic tumor antigens in surrendered immunity and the significance of autologous tumor cell-based vaccination in precision medicine” the authors discuss the utility of autologous cancer vaccines, the reprogramming of tumor antigenic expression, combinational anti-tumor therapies, and their relevance to the precision medicine. Below are the comments to improve the manuscript.

1. Lines# 206-207: CD4+CD25+FoxP3+ tumor-derived regulatory T cells (Tregs) play central roles in immune suppression. The + sign denoting the expression of markers should be superscripts throughout the manuscript.

Response: Thanks for the significant comments from the Reviewer. We checked carefully and superscripted the “+” sign throughout the manuscript.

2. Lines# 386-388: CB17-Prk- 386 dcscid/NcrCrl (NOD.SCID) mice for live tumor conservation (named as XCTVT, X: xenogeneic). A thorough spelling check should be carried out throughout the manuscript.

Response: Thanks for the significant comments from the Reviewer. We corrected the misspelling and checked the words throughout the manuscript.

3. The authors should include diagrams to depict their take-home message/hypothesis.

Response: A diagram illustrating the hypothesis in this study was involved in the revised manuscript (Lines 518 - 530).

Reviewer 2 Report

Ke et al. submitted an interesting review summarizing the current knowledge concerning tumor-based vaccinations and the molecular background of cancer immunotherapy. The manuscript is generally well-written and clearly shows the complex correlation between immunology and cancer development. Included figures illustrate the content and are helpful in their understanding. Only minor issues appeared during manuscript revision:

  1. It would be advantageous to include in the 5.3. section a table summarizing the clinical trials focused on cell-based vaccines (vaccine, indication, molecular target, NCT number, short comment). That would give the full view of the tumor cell-based vaccines in the current anticancer strategies. 
  2. The References section should be correct according to the MDPI template. All Journals' names should be abbreviated (correct: 3, 5, 16, 17, 19, 21, 23, 34, 35, 46, 47, 53, 55, 56, 59, 64, 79, 87, 92, 96, 98, 101, 110, 114, 115, 122, 123, 125, 128, 133, 136, 138, 139, 142, 149, 151, 152, 158, 163, 164, 165, 166,
  3. ref 54 - Cell Rep (nor rep)
  4. ref 76 - FEBS J (not Febs j)
  5. ref 117 - EMBO J (not Embo j)

Author Response

Ke et al. submitted an interesting review summarizing the current knowledge concerning tumor-based vaccinations and the molecular background of cancer immunotherapy. The manuscript is generally well-written and clearly shows the complex correlation between immunology and cancer development. Included figures illustrate the content and are helpful in their understanding. Only minor issues appeared during manuscript revision:

It would be advantageous to include in the 5.3. section a table summarizing the clinical trials focused on cell-based vaccines (vaccine, indication, molecular target, NCT number, short comment). That would give the full view of the tumor cell-based vaccines in the current anticancer strategies.

Response: Thanks for the significant comments from the Reviewer. We added a Table to illustrate the selected clinical trials of cancer cell-based vaccines (Lines 483 and 504 - 507).

The References section should be correct according to the MDPI template. All Journals' names should be abbreviated (correct: 3, 5, 16, 17, 19, 21, 23, 34, 35, 46, 47, 53, 55, 56, 59, 64, 79, 87, 92, 96, 98, 101, 110, 114, 115, 122, 123, 125, 128, 133, 136, 138, 139, 142, 149, 151, 152, 158, 163, 164, 165, 166,

ref 54 - Cell Rep (nor rep)

ref 76 - FEBS J (not Febs j)

ref 117 - EMBO J (not Embo j)

Response: We carefully checked the Reference section the Reviewer mentioned. The cited journals have been corrected and abbreviated according to the MDPI template (ISO 4 rules) except for ref. 56, 79, 96, 98, 101, 114, and 149. These journals are Oncoimmunology (ref. 56, 79, 96, and 98), Cell (ref. 101), Histopathology (ref. 114), and Immunotherapy(ref. 149), of which the Journal names are the same as their abbreviations. We thank the correction that improved our work from the Reviewer.